# Tree Willow Root Growth in Sediments Varying in Texture

**Ian McIvor [1],\* and Valérie Desrochers [2]**

[1]  Sustainable Production Group, The New Zealand Institute for Plant & Food Research Limited, Private Bag 11600, Palmerston North 4442, New Zealand
[2]  Institut de recherche en biologie végétale, Montreal, QC H1X 2B2, Canada; valeriedesrochers26@gmail.com
\*  Correspondence: ian.mcivor@plantandfood.co.nz; Tel.: +64-6-9537673

**Abstract:** We investigated the early root development of *Salix nigra* L. willow grown from cuttings in the different riverbank sediments; silt, sand and stones. Cuttings were grown for 10 weeks in layered sediment types in five large planter boxes, each box having three separate compartments. The boxes differed in the proportion of silt, sand and stones. At 10 weeks, the roots were extracted and sorted into diameter classes (≥2 mm; 1 < 2 mm; <1 mm) according to sediment type and depth. Root length and dry mass were measured and root length density (RLD) and root mass density (RMD) calculated. Root development of *S. nigra* cuttings varied with the substrate, either silt, sand or stones. Roots initiated from the entire length of the cutting in the substrate but with a concentration of initials located at the bottom and close to the bottom of the cutting. There was substantial root extension into all three substrates and at all depths. Generally, RMD was higher in the stones, influenced by having the bottom of the cuttings in stones for four of the five treatments. RMD was highest for roots <1 mm diameter. RMD of roots <1 mm diameter was least for those roots growing in sand. Whereas RLD for roots >0.5 mm diameter was highest in the sand, RLD of roots with diameter <0.5 mm was lowest in sand. Roots of *S. nigra* cuttings were least effective in binding sand, primarily because of low RLD of roots <0.5 mm diameter. It is surmised that sand lacks water and nutrients sufficient to sustain growth of fine roots compared with silt and even stones. RLD for roots >0.5 mm diameter was lowest in silt likely due to the greater resistance of the substrate to root penetration, or possibly the greater investment into smaller roots with absorption capability.

**Keywords:** coarse root; fine root; cutting; silt; sand; stone; *Salix nigra* L.

## 1. Introduction

In New Zealand, live willows, primarily male tree willows, are the most important biotechnical tool for frontline protection of riverbanks from water erosion, particularly in periods of high flow. Willows are not native to New Zealand and there is considerable sensitivity to seed spread from planted willows along waterways, hence male willows are preferred. Planting cuttings from known parent trees reduces risk of seed dispersal. Tree willows used for river engineering in New Zealand are established from large cuttings up to 3.5 m long and 70 mm in diameter. Any threat to willow health and effectiveness in binding sediment and stabilizing river banks is also a threat to property, livelihoods and possibly human lives. During the period 1997–2003 willow sawfly (*Nematus oligospilus* Forster) posed such a threat [1–3], with willows along rivers in some parts of the country experiencing severe and repeated defoliations and some mortality from the feeding activity of willow sawfly larvae. Since its arrival in New Zealand in 2014, giant willow aphid (*Tuberolachnus salignus* Gmelin) has posed a significant threat to willow health [4]. The threat of this aphid to biotechnical effectiveness of willows in protecting riverbanks is as yet unclear. This incursion arises at the same time the country

is facing possible increased rainstorm frequency and flood risk consequent on climate change [5]. The failure of willows to protect bank sections along the Whakatane River in Bay of Plenty, New Zealand during a severe flood in April 2017 was reported by river engineers, who prompted questions about whether the effectiveness of the willow root system was compromised by the feeding activity of giant willow aphid (GWA). Potentially a reduction in stored carbohydrate in the root system due to feeding activity of GWA could reduce the production of fine roots that bind to sediment and raise biotechnical effectiveness [6–8]. Carbohydrate storage in non-absorptive roots plays an important role in maintaining tree survival after the termination of photosynthate flow from above ground sources [8].

Riverbank sediments get transferred and deposited during floods in layers according to particle size with gravels, stones and boulders being overlaid with sand and with silt dominant in the top layer [9]. In the absence of binding agents, such as tree roots, fine particles are highly mobile in floods, the mean erosion rate increasing with deposit height, deposit width and decreasing grain size [10]. *Salix* spp. are noted as being effective in binding sediment and are not easily washed away with eroded sediment [11].

Published quantitative data on normal root development of tree willows growing along riverbanks is scarce, though other studies have investigated willow root development in different soil types [12–14]. This could be considered of little consequence as willows are seen in practice to be generally effective in binding sediment under flood conditions, with any failings understood as due to the power of the floodwater. However, determining the effect of a pest organism on a tree willow root system cannot be determined without first knowing what a healthy root system is like. Hence, this investigation was designed to first identify how a normal tree willow root system develops.

*S. nigra* is a colonising floodplain species that produces a massive root system and stabilises streambank sediments [15]. *S. nigra* was used in this experiment as a model tree willow to assess root development from a cutting (the propagating material used for riverbank stabilization in New Zealand) in the different kinds of substrates (stones, sand, silt) found on riverbanks and in varying proportion. We hypothesised that willow root morphology will differ with the sediment type, and contend that knowledge of the differences will contribute to understanding factors that may result in willow failure on riverbanks.

## 2. Methods

To investigate tree willow root development in river bank sediments a box trial was set up in a research facility at The New Zealand Institute for Plant & Food Research Limited (PFR), Palmerston North. Five plywood boxes (dimensions 1.6 m × 0.8 m × 0.4 m) were constructed, each box having three equal compartments (dimensions 0.5 m × 0.8 m × 0.4 m) separated by plywood bracing. The bracing had a central hole 0.6 m × 0.3 m allowing roots to extend naturally. To observe the roots during the experiment, the plywood on one long side of the boxes was replaced with transparent polycarbonate. Fifteen 0.9 m cuttings taken from two year old shoots of *S. nigra* were cut fresh, pre-soaked in a bucket of water till emerging root primordia were visible, and planted in the boxes in November 2017, three cuttings per box, each cutting in its separate compartment. Cuttings were of similar thickness with top diameter averaging 29.6 mm and basal diameter 46.3 mm. They were then located in an ambient environment. Drainage holes were covered with a fine wire mesh to allow drainage while retaining the substrate. Each box was attributed different proportions of silt, sand and "river run" stones sourced from a local quarry, representative of the river bank variation along a river course (stones phase, mixed silt/sand/stones phase, silt/sand phase). Different substrate proportions were allocated to each box (Table 1). The same proportion of substrate was replicated in each of the three compartments in a single box. Ten percent of volume was marked every 7 cm up the side of the box. A 60 mm diameter polyvinyl chloride pipe was centered into each compartment, extending to the bottom of the compartment, and substrate filled around it to 0.7 m depth, substrate by substrate (Table 1). The 0.9 m *S. nigra* cutting was inserted into the pipe and the pipe removed allowing the

substrate to close against the cutting to a depth of 0.7 m without unduly disturbing the substrate layers, leaving 0.2 m of cutting above ground. The media settled as the experiment progressed.

**Table 1.** Arrangement of the sediment layers in each box.

| Layer % (from top) | Depth cm | Box 1 | Box 2 | Box 3 | Box 4 | Box 5 |
|---|---|---|---|---|---|---|
| 0–10 | 0–7 | | | | | silt |
| 10–20 | 7–14 | | | silt | silt | |
| 20–30 | 14–21 | | silt | | | sand |
| 30–40 | 21–28 | silt | | | | |
| 40–50 | 28–35 | | | sand | | |
| 50–60 | 35–42 | | | | sand | |
| 60–70 | 42–49 | | | | | stones |
| 70–80 | 49–56 | | sand | | stones | |
| 80–90 | 56–63 | sand | | stones | | |
| 90–100 | 63–70 | | stones | | | |

The cuttings were provided with natural rainfall, supplemented with water supplied by overhead irrigation twice daily once the foliage emerged. No additional nutrients were added to the mix.

Ten weeks after the cuttings were planted the experiment was terminated. One side of the box was removed and the roots were extracted manually from each compartment in turn. A tarpaulin was employed to collect any substrate that fell from the box and to capture all roots.

Of the substrates, the silt and sand were gently removed from each compartment using a spade, in 7 cm depth intervals (Table 1), to keep the root system as intact as possible and sieved (10 mm mesh) to collect the roots. Where stones were included as a substrate in a compartment (all except those in box 1), they were treated as a single sample, varying in depth (Table 1) because their large size made separation impossible. A total of 100 substrate samples in total were collected from the 5 boxes. The roots in each sample were separated from their substrate and washed, their diameter measured with calipers, then cut and allocated to three diameter classes (≥2 mm; 1 < 2 mm; <1 mm) [16]. Roots in diameter class ≥1 mm are subsequently described as coarse roots. For each box, compartment, cutting, substrate type, depth, root length (RL) (measured with a ruler to the nearest cm) and root mass (RM) were recorded. All root diameter classes in each sample were separately oven-dried at 70 °C for 48 h, weighed to 0.01 g and RM recorded. RL (m) of the two root diameter classes ≥1 mm was measured. For root diameter class <1 mm, RL of roots with diameter 0.5 < 1 mm was measured and recorded for a random subset only of the samples from each substrate because of practical constraints. For root diameter class <1 mm, we calculated a relationship between the RL of roots with diameter 0.5 < 1 mm and RM of the samples for each substrate, assumed this to be a constant relationship for all the samples not measured, and so calculated RL for root diameter class 0.5 < 1 mm from RM for the remaining unmeasured samples. The mean density of RM and the total RL were calculated for each substrate in the different depth intervals. The root length density (RLD) and the root mass density (RMD) were calculated for the various substrates at each depth interval (7 cm except for stones) and to 0.70 m depth for all the root diameter classes. All shoots emanating from the cutting were counted, regardless of size. Shoot dry mass for each cutting was measured and recorded. The top and bottom diameters of the cuttings were measured at the beginning and the end of the experiment.

Root data were analysed for significant ($p < 0.05$) differences between substrate types and diameter using ANOVA (Genstat 17th Edition; VSNi Limited, Hemel Hempstead, UK, 2014). RLD and RMD data were log transformed to stabilize variance, and a mixed effects model was fitted, with fixed effects for substrate and diameter, and random effects for box and plant within box. Roots 10–20 mm and over 20 mm were combined into one category, as roots over 20 mm were very rare. Raw data are presented in the tables and figures.

## 3. Results

### 3.1. Description of the Roots

The live roots observed through the polycarbonate were a mixture of fine and coarse roots. Some roots were white (not lignified) and others golden brown (lignified). New roots were white (Figure 1).

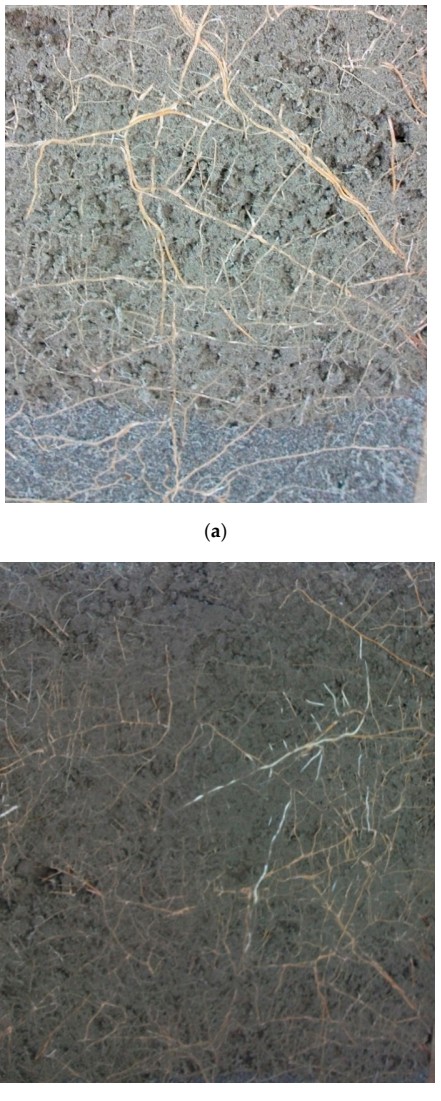

(**a**)

(**b**)

**Figure 1.** (**a**) Contrasting root growth in silt and sand layers. Note the greater number of very fine roots in the silt. (**b**) Characteristic root development in silt dominated by very fine roots and high production of new roots visible as white roots.

Roots developed from the whole buried length of the cutting, with the greater amount of root development coming from the bottom of the cutting. The finest roots in the stones often adhered to the stones, their growth following the various indentations of the stones where fine sediments were often present. Roots were easiest to recover from the sand, followed by the stones and then silt.

The roots grew in all directions in the substrates. The roots originating in the sand layer tended to grow fine roots upward into the silt. We observed a higher presence of finest roots (roots < 0.5 mm diameter) in the silt than the sand (Figure 1).

### 3.2. Shoot Production in the Five Boxes

Total shoot dry mass for the cuttings in the five boxes varied (Table 2). Shoot mass of individual trees ranged from 36 g to 128 g, with an overall mean mass of 71.6 g. Except for the trees in box 1, shoot numbers were similar in each box (Table 2). The mean overall was 12.1 shoots per tree for the 15 trees. Differences in shoot number ($p = 0.064$) and shoot dry mass ($p = 0.098$) between plants in the different boxes were not significant. The cuttings in the high silt substrate (box 1) had near-significantly more shoots than the cuttings in the other boxes.

**Table 2.** Growth in cutting diameter and shoot biomass during the experiment.

| Box | Substrate | Cutting Top Diameter (mm) | | Cutting Basal Diameter (mm) | | Shoots | | Cutting Volume (cm$^3$) | Total Root Mass (g) | Coarse Root Length (m) |
|---|---|---|---|---|---|---|---|---|---|---|
| | Silt:Sand:Stones | Start | End Mean (SE) | Start | End Mean (SE) | Mean No. (SE) | Mean DW (g) (SE) | Mean (SE) | Mean (SE) | Mean (SE) |
| 1 | 80:20:0 | 30.7 | 31.8 (2.5) | 47 | 47.7 (3.1) | 16.0 (1.0) | 107 (13) | 1128 (151) | 35 (3) | 31 (3) |
| 2 | 60:30:10 | 28.3 | 28.7 (2.9) | 44.3 | 47.2 (4.6) | 11.3 (1.5) | 68 (22) | 1036 (206) | 26 (4) | 32 (5) |
| 3 | 30:40:30 | 30.0 | 30.7 (2.9) | 47 | 50.2 (4.1) | 11.3 (1.9) | 73 (13) | 1171 (193) | 32 (2) | 45 (1) |
| 4 | 20:30:50 | 29.7 | 30.3 (1.2) | 48.7 | 49.3 (2.6) | 11.7 (0.9) | 47 (7) | 1127 (106) | 24 (3) | 11 (3) |
| 5 | 10:20:70 | 28.0 | 30.2 (2.2) | 45.3 | 47.0 (1.2) | 10.3 (0.7) | 63 (8) | 1056 (89) | 34 (8) | 10 (3) |

Differences in total root mass between boxes was not significant ($p = 0.336$). However, differences in coarse RL between boxes was significant ($p < 0.001$; LSD = 10; B3 > B1, B2 > B4, B5).

### 3.3. Root Distribution within the Five Boxes

Coarse RL was greater in the boxes with the greater proportion of silt (Table 3; F = 20.9 on 4 and 10 df. $p < 0.001$; LSD = 10) and greatest in the box with the highest proportion of sand. The two boxes (4, 5) with a low proportion of silt and a high proportion of stones had the lowest coarse RL. Root mass was greatest in the box with the highest proportion of silt. Root mass was high in the box with the greatest proportion of stones, contributed by a large mass of roots at the bottom of the cuttings. However, with fine roots (<1 mm diameter), the % RM was in similar proportion to the silt present in the box, whereas for sand the finest RM fraction was underrepresented. For example, in box 2, 65% of fine RM was found in silt (60% of the sediment by volume), whereas 18% was found in sand (30% by volume). Fine RM was overrepresented in the stones, contributed by the greater number of roots initiating from the bottom of the cutting.

**Table 3.** The proportion of root length (RL; roots with diameter ≥1 mm) and root mass (RM) of coarse roots (diameter ≥1 mm) and fine roots (diameter <1 mm) found in the different proportions of sediment in each box.

| Box | Sediment | % | % RL ≥ 1 mm Diameter | % RM ≥1 mm Diameter | % RM <1 mm Diameter |
|---|---|---|---|---|---|
| 1 | silt | 80 | 57 | 58 | 83 |
| | sand | 20 | 43 | 42 | 17 |
| | stones | 0 | 0 | 0 | 0 |
| 2 | silt | 60 | 42 | 29 | 59 |
| | sand | 30 | 34 | 32 | 28 |
| | stones | 10 | 24 | 39 | 13 |
| 3 | silt | 30 | 16 | 13 | 35 |
| | sand | 40 | 44 | 37 | 39 |
| | stones | 30 | 39 | 50 | 26 |
| 4 | silt | 20 | 10 | 1 | 23 |
| | sand | 30 | 23 | 37 | 27 |
| | stones | 50 | 66 | 62 | 50 |
| 5 | silt | 10 | 4 | 2 | 8 |
| | sand | 20 | 6 | 18 | 8 |
| | stones | 70 | 90 | 80 | 84 |

### 3.4. Root Density

The mixed effects model indicated significant main effects for substrate and diameter (F = 22.3 on 2 and 185 df. $p < 0.001$, and F = 259.7 on 2 and 185 df. $p < 0.001$, respectively) but no significant interaction with plant or box (F = 0.2 on 4 and 184 df. $p = 0.947$). Mean RLD of coarse roots was significantly lower in silt ($p = 0.03$) than in sand and stones (Figure 2). Coarse root RLD was higher in sand than stones, RLD of fine roots 0.5 < 1 mm diameter was highest in sand (264 m m$^{-3}$), close to three times higher ($p = 0.04$) than in silt (94 m m$^{-3}$).

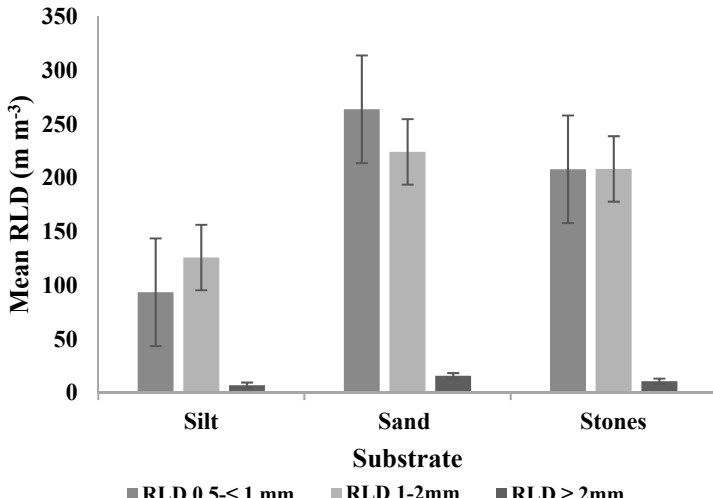

**Figure 2.** Variation in root length density (RLD) with depth class and substrate. Error bars are 1 SE.

RLD for coarse roots generally increased with depth, whereas for the fine root diameter class (0.5 < 1 mm), RLD was highest in the upper layers in both silt and sand (Figure 3). Differences in RLD with depth were not significant. The effect of depth (but not diameter × depth) was significant, although comparatively small relative to the diameter and substrate effects (F = 3.6, $p < 0.001$).

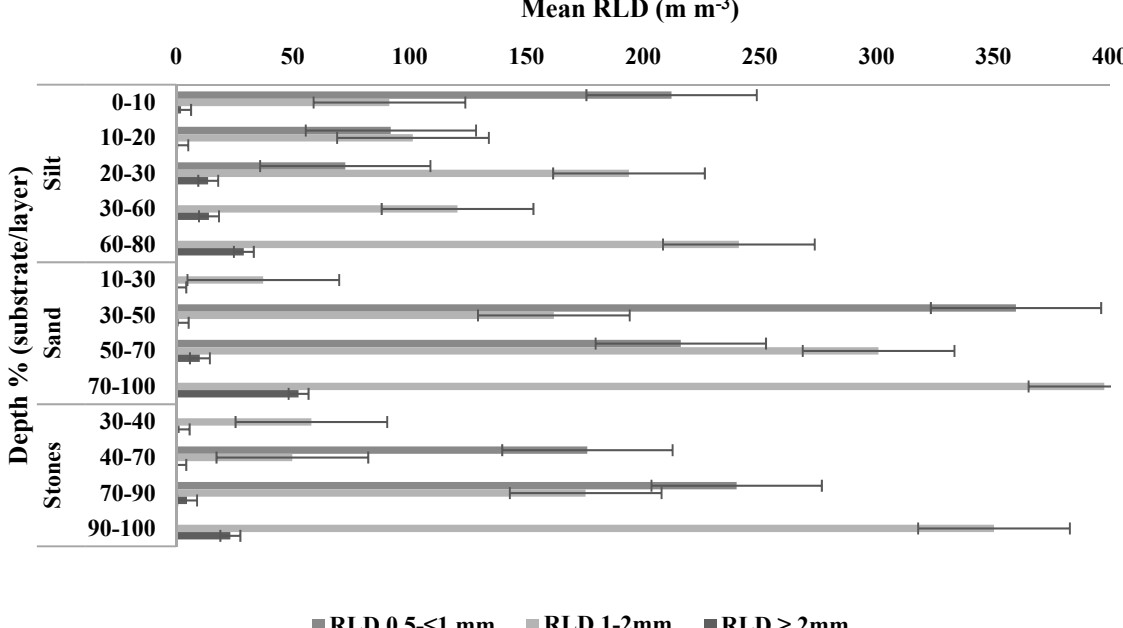

**Figure 3.** Variation in root length density (RLD) with depth in the three substrates. Error bars are 1 SE. Layers missing a value for the 0.5 < 1 mm root diameter were not sampled.

The mixed effects model indicated significant main effects on RMD for substrate and diameter (F = 36.6 on 2 and 265 df. $p < 0.001$, and F = 506.1 on 2 and 261 df. $p < 0.001$, respectively) but no significant interaction (F = 1.3 on 4 and 261 df. $p = 0.286$). Mean RMD was greater in stones (F = 36.6 on 2 and $p < 0.001$) than in sand or silt. Mean RMD was higher in sand than in silt but not significantly so. Mean RMD was higher for fine roots than for coarse roots in each substrate; and was highest in the stones (254 g m$^{-3}$) and lowest in the sand (135 g m$^{-3}$) (Figure 4). Coarse root RMD was highest in stones, intermediate in sand (Figure 4) and significantly lower in silt ($p < 0.001$).

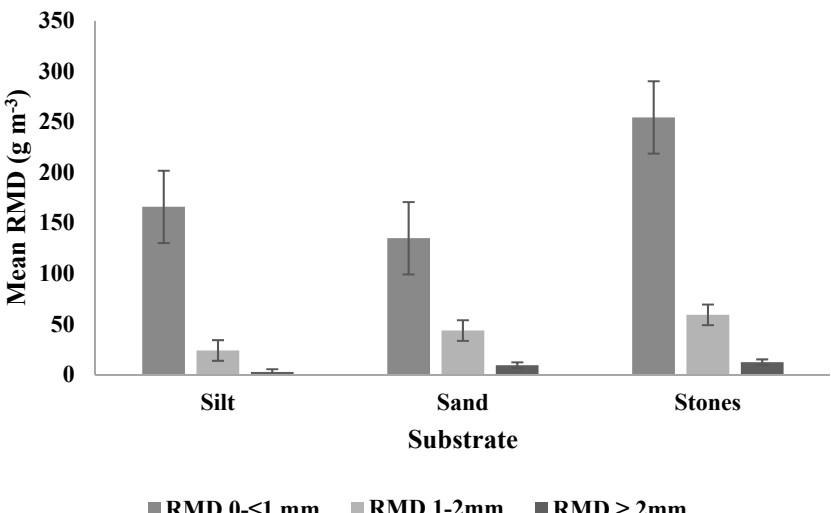

**Figure 4.** Mean root mass density (RMD) separated by diameter class and substrate. Error bars are 1 SE.

RMD generally increased with depth in each substrate for all root diameter classes (Figure 5), with the highest density in the deepest layer. Fitting the mixed effects model to the log of the RMD, depth (and diameter × depth) were significant, although comparatively small relative to the substrate and diameter effects (F = 6.1, $p < 0.001$ and F = 1.8, $p = 0.019$ for depth and depth × diameter).

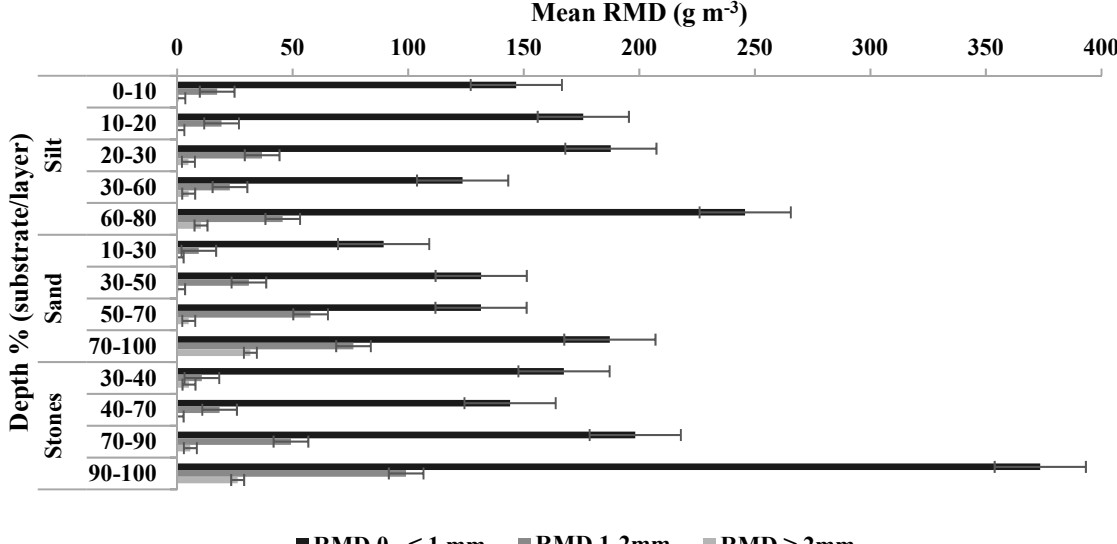

**Figure 5.** Variation in root mass density (RMD) with depth separated by diameter class and substrate. Depth scale is % (each 10% interval = 7 cm depth from the surface). Error bars are 1 SE.

Mean RMD of roots <1 mm diameter was 246 g m$^{-3}$ in the silt, 187 g m$^{-3}$ in the sand and 374 g m$^{-3}$ in the stones.

### 3.5. Comparing Root Production in Each Box

The ranking for mean RMD of all root diameter classes was similar in each box; higher RMD of finest roots, followed by the 1 > 2 mm roots, then the ≥2 mm roots. The mean RMD of root classes differed between boxes; the finest (<0.5 mm) root RMD was greatest in box 5 with the highest proportion of stones (261 g m$^{-3}$), and lowest in box 3 with the highest proportion of sand (139 g m$^{-3}$), while the 1 < 2 mm diameter were greatest in box 3 (66 g m$^{-3}$) and lowest in boxes 4 (20 g m$^{-3}$) and 5 (21 g m$^{-3}$). However, the RMD of thicker roots was greatest in box 1 (15 g m$^{-3}$) and lowest in box 4 (2 g m$^{-3}$).

Total RM was highest in the box with the greatest volume of stones (box 5) followed by the box with the highest silt content (box 1). However, root production varied sufficiently between the trees within each box that the differences in total RM between boxes were not significant.

For the subset of samples for which root length (RL), root mass (RM) and contribution (%) of samples from ≤0.5 mm and between 0.5–1 mm size diameter categories were determined, mean RM contribution of roots 0.5 < 1 mm was 6%, and that of roots ≤0.5 mm was 94%. Mean RL for 0.5–1 mm roots was 1.31 m, and for roots ≤0.5 mm was estimated at 11.8 m following a previous approach [12]. However, these data are more representative of the situation for roots in silt (11 samples) because fewer samples were analysed for sand and stones.

Specific coarse root length (SRL) differed between substrates, being lower ($p < 0.001$) in stones (3.05 m g$^{-1}$) than in sand (4.94 m g$^{-1}$) or silt (5.00 m g$^{-1}$). For fine roots of 0.5 < 1 mm diameter, SRL did not differ between substrates with a range of 9.54–11.52 m g$^{-1}$. SRL for roots < 0.5 mm diameter was calculated at between 86–104 m g$^{-1}$.

## 4. Discussion

Field studies on willow roots in New Zealand focused on larger roots with diameters of ≥2 mm because of the time involved and the challenge of extracting both large structural and fine roots in a single operation [17,18]. It was demonstrated that tree willow with roots growing in recently deposited sandy silt soil grew at a much faster rate than tree willow grown in pastoral soils and that root extension occurred at a much faster rate also [18]. A pot study [12] investigated tree willow fine and coarse root

development, but in pastoral soils rather than river sediments. Consequentially, comparisons with these studies will not be made, but there will be comparisons made with aligned studies.

Root length (mass) density, the length (mass) of roots per unit volume of soil, is one of the important parameters required to understand plant performance [19], in this case, ability to bind sediments of various types. In river sediments, unlike other soils, a very high percentage of the roots can be recovered, including the finest roots. RLD for roots >0.5 mm diameter was higher in the sand than the other two substrates. However, the lower RMD for roots <1 mm diameter in sand compared with silt and stones indicates less of the finest roots (<0.5 mm diameter) are present in sand layers. These are the roots that function to absorb water and nutrients. Sand allows easier passage for root extension, however sand does not appear to provide water and nutrients to the same extent as the other two substrates. Roots were able to be separated from the sand relatively easily compared with silt and stones, primarily because of the very low quantities of the finest roots and little adhesion to the sand. As a consequence of this differential root development in sand, we predict that willows will be least effective in binding sand and reducing erosion of sand in floods. RLD for roots >0.5 mm diameter was lowest in silt likely due to the greater resistance of the substrate to root penetration, or possibly the greater investment into smaller roots with absorption capability. Roots were harder to separate from silt than from the other two substrates because of both high fine root densities and silt particles bound to the roots.

A study investigated root reinforcement by fine roots <2 mm diameter for *S. nigra* trees in two creeks in Mississippi, USA, subject to regular flooding and differing particle size distribution (Sardis creek 20% sand, 60% silt and 20% clay; Goodwin creek 61% sand, 26% silt and 13 % clay) [20]. Soil moisture contents for the two soils were 17% (Goodwin creek) and 18% (Sardis). The mean tensile load required to pull *S. nigra* seedlings (mean stem diameter 7.98 mm at Sardis and 7.29 mm at Goodwin creek) was 133.6N at Goodwin and 523.9N at Sardis [20]. Failure at the site with the coarser sediment (Goodwin) was mostly (68%) because the entire root system pulled out of the soil body, whereas at Sardis, 100% of root system failures occurred due to fracture of the larger roots. The finest roots have the capacity to keep soil particles together, and so are effective at soil binding. The nature of substrate, by the predominance of fine particles, will influence the roots–particles association [15]. Likewise, the larger particle sizes of sand did not favour the proliferation of fine roots and did not prevent the removal of particles during flood events [20]. Sand erosion will be best protected when the sand is overlaid or well mixed with silt, which contains a high proportion of those fine roots. Fine roots release exudates that cement particles together, form aggregates, bind the upper layer and protect the sand. The study [20] and this present study strongly suggest that for banks protected by willow trees, the risk of river and stream bank erosion will increase when a higher proportion of sand is present in the soil because of the lower fine root proliferation. While Collins did not excavate roots to discover how much fine root was present in each soil type, our observations confirmed her hypothesis that fine root presence was lower in soils with a high sand content. Fine roots develop the three dimensional nature of root occupancy enabling the root system to better resist wrenching forces arising from different directions. Continuous depth models used to describe tree vertical root distribution for soil based trees [18,21] are not suitable for riverine sediments. The nature of the sediment and moisture availability provide better explanations for riverbank root distribution [15,22]. Schaff and colleagues showed that *S. nigra* pole growth along a riverbank in Mississippi, USA, particularly in the second year of growth was markedly higher (14% increase in biomass) in sandy sediments than in silt/clay sediments, and was lowest where the water table was deepest.

Our findings suggest that proliferation of fine roots is also a characteristic of a stony substrate, however these findings need to be qualified. A large number of roots emerged from the bottom of the cuttings growing in gravel in four of five boxes. This response may be more characteristic of *S. nigra* than other tree willows [12]. However, there was also a large amount of fine root in a substrate offering low resistance to root penetration and a high and constant moisture content. A previous study of root development in *Salix matsudana × alba* 'Moutere' tree willow growing along the Hutt river,

New Zealand showed that root development in stones/gravel can be almost nil when the gravel has no water present [23].

We used a mixture of substrate similar to what may occur on a river bank, some having a lot of stones and others composed largely of silt/sand. There were differences in root development between boxes that could be related to different proportions of substrates, such as occur at different reaches of a river. These findings enable some prediction about how the root network will develop as the tree grows, where the greatest pressure on it will come if there is a flood, i.e., which substrate layer is the most mobile and which one is the hardest. The sand had the lowest RMD of fine roots and fell apart so readily when we were extracting roots that we presume the sand will offer low root system anchorage. It will be more easily eroded by forces associated with increased water flow than the other sediments, and will create weak points in the profile amplifying the erosion. Root proliferation was also occurring from the bottom of the cuttings or pole. This strengthens the anchorage of the trees and increases the effectiveness of the shallow roots by reducing mechanical pressure. If the deep roots can absorb the large mechanical forces exerted on the trunk of the tree by high water flows and accompanying debris, then the shallower roots are better positioned to resist the smaller erosive forces. The tree is likely to be better anchored in flood conditions if roots are able to extend into a gravel or boulder layer which will require greater forces to shift than sand or silt. This will be promoted by burying cuttings down to this layer at planting.

While this report has focused on the role of willow roots in stabilising river and stream banks, it can be observed that the root system in sediments is well developed right to the water table and ideally positioned to intercept nutrient runoff from adjoining land before it enters the aquatic system, and that this root system can be developed faster at depth by planting large cuttings to appropriate depths, e.g. water table.

Findings of this study will contribute to understanding possible reasons for tree willow failure during floods, and also the effects of such pests as willow sawfly and giant willow aphid on tree willow small and fine root production.

## 5. Conclusions

Root development of *S. nigra* grown from cuttings varied with the riverine substrate, either silt, sand or stones. Roots initiated from the entire length of cuttings in the substrate but with a concentration of initials located at and close to the bottom of the cutting. Generally, RMD was higher in the stones. The higher RMD in the stones was influenced by having the bottom of the cuttings located in stones for four of the five treatments. RMD was highest for roots <1 mm diameter. Whereas RLD for roots with >0.5 mm diameter was highest in the sand, RLD of roots with a diameter of <0.5 mm was lowest in sand. RLD for roots >0.5 mm diameter was lowest in silt, likely due to the greater resistance of the substrate to root penetration, or possibly the greater investment into smaller roots with absorption capability in the substrate with higher nutrient and water content. Roots of *S. nigra* were least effective in binding sand, primarily because of low RLD of roots <0.5 mm diameter. It is surmised that sand lacks water and nutrients sufficient to sustain growth of fine roots compared with silt and even stones.

**Author Contributions:** I.M. was responsible for conceptualization, methodology, validation, resources, formal analysis, writing—review and editing, project administration; V.D. was responsible for methodology, investigation, data curation, formal analysis , writing—original draft preparation.

**Funding:** This research was funded by Ministry of Business, Innovation & Employment, Project P/442060/10.

**Acknowledgments:** Technical assistance was provided in setting up and maintaining the experiment and data collection.

**Conflicts of Interest:** The authors declare no conflict of interest. The funders had no role in the design of the study; in the collection, analyses, or interpretation of data; in the writing of the manuscript, or in the decision to publish the results".

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
