# Peer review of "Tree Willow Root Growth in Sediments Varying in Texture"

_forests, doi:10.3390/f10060517_

Round 1

Reviewer 1 Report

General comments: The topic of this study is important and warrants investigation to aid soil conservation in forested waterways.  However, this paper does not contain the level of detail needed to convey the methods and outcome of an experiment.  Of primary concern is the Methods section which is incomplete because it does not contain the level of detail needed to understand what was done.  Furthermore, non-scientific descriptive terms are used throughout the paper when established and referenceable terminology is available. Also, the experimental design and/or model used for ANOVA are not described, and based on the information given, it is questionable whether this study is statistically sound.  It appears that the study is based on observations from five Salix cuttings with one cutting per substrate combination.  This does not appear to be replicated.  My review comments terminated after the Methods section.

At the least, the paper would need a very thorough revision with assistance to make sure the necessary information was included in the Methods section. However, lack of evidence of replication suggests that it is not publishable in a refereed journal.

Specific comments:

Introduction

I believe that specific epithets are needed after all scientific names.

Page 2, line 51: This statement needs expansion with what the “resources” are (carbohydrate, mineral nutrients, etc.).  There should also be some references documenting that these “resources” are actually available for root growth based on others’ research.

Page 2, line 57: State what you are investigating in scientific terms (e.g., morphology, phenology). What it “is like” is very vague not scientific.

Page 2: The last two paragraphs of the Introduction seem out of place because they follow your statement of purpose (lines 58-59). A suggestion is to move lines 60-65 to line 53.

Page 2: You need a final paragraph of the Introduction containing your hypothesis and objective. A suggestion is (1) Last paragraph stating with sentence at lines 68-69. (2) Next sentence is lines 66-68. (3) Next sentence is your hypothesis. (4) Last sentence is lines 58-59.

You need to justify the use of cuttings as a surrogate for naturally seeded willow.

Methods

Page 2, paragraph 1: More information is needed about the source of the cuttings.  For example, what was their source (e.g., native shoots from mature, naturally growing trees), what age were the trees at the time of cutting, and when were the cuttings made.

Page 2: You probably need an entire paragraph to describe the cuttings (see above).  Another paragraph is needed to describe the boxes.  The dimensions of the box are given, but you need to add dimensions of the rooting zone, and the three compartments serving as replications. This paragraph should contain information on how the rooting medium was created and how the PVC pipe was used.  I cannot follow how you centered the PVC pipe in each box, but there were three compartments in each box.  You need to provide more detail such as that the PVC pipe depth reached to bottom of the box.  Then start another paragraph with information on how the cuttings were planted (depth) in the PVC pipe and how and when the pipe was removed.  So there were only 5 cuttings used in the entire study?

Page 2: You mention that the boxes are near an irrigation system, but then you state that a sprinkler was used to water the cuttings.  It would be clearer if you used one term for these attributes.

Page 2, line 73: Is “root buds” a real term?  If not, try to use the technically correct term for this, perhaps “emerging root primordia”.

Page 2, line 85: Was the side of the box covered with transparent polycarbonate, or was an observation window created by replacing the wood on one side of the box with transparent polycarbonate?

Page 3, line 91: A new paragraph should begin with “Ten weeks…”  Statements like “layer by layer” are not appropriate in a scientific document. You need to do a better job of describing the depths of the “layers” earlier in the Methods.  Then at this point, make a statement indicating that all roots were extracted from each 7 cm layer substrate.  

Page 3, line 97: What sieve size?

Page 3, line 97: Samples of what… substrate, roots?  What was the volume of the 100 samples? Why wouldn’t you sample all roots? How does the reader know that the samples you chose were representative of those in the box? 

Page 3, line 99: What was the basis of these diameter classes? Include a reference.

Page 3, line 100: Here you refer to “cell.” Are these the three compartments per box described earlier?  If so, use one term only. Was it 100 samples per cell or box?

Page 3, line 100: What is a “sample box”?  Is this the same as “box”?

Page 3:  How did you measure root length?

Page 3, lines 96-110: I cannot tell what was done with the harvested roots.  I can only guess at this point.  If I guess, then this paragraph should be separated into several with one each for (1) root harvest by layer, the tools used, root class identification, live/dead identification, (2) dry mass determinations of 100% of the root harvest by layer, how measured (nearest 0.1 g?), (3) 100 subsamples (by what…volume, fresh weight) from each cell for RL determinations, how RL determined, (4) relationship development, equations by root class, (5) What RLD is and how is was calculated by cutting, (6) Shoot variables.

Page 3: There is no description of the experimental design, or how you did your mean separations (Figure 2).  As it stands, it looks like 5 cuttings were planted in each of five boxes. Each box had a specific substrate combination.  I don’t detect any replication except among the three cells per box. But, I’m not sure if this qualifies as replications since the observations came from the same cutting and box. 

Results

Page 3, line 113: “thin and thick roots” are not scientific terms.  You need to search the literature for standard descriptive terms that can be applied to your study.  You must define your root categories based on the literature.

Page 4, line 130: Are these cuttings or trees?  I don’t think they should be called trees. Why do you start with shoot description when root description is the emphasis of your paper?

Page 4: ANOVA tables should be presented.

Page 4, line 131: There is no indication of how shoot number was quantified. What qualified as a shoot?

Page 6-7: Bars in figures are too difficult to resolve.

Author Response

Introduction

I believe that specific epithets are needed after all scientific names.

Page 2, line 51: This statement needs expansion with what the “resources” are (carbohydrate, mineral nutrients, etc.).  There should also be some references documenting that these “resources” are actually available for root growth based on others’ research. I have changed resources to carbohydrate

Page 2, line 57: State what you are investigating in scientific terms (e.g., morphology, phenology). What it “is like” is very vague not scientific. Changed to how a willow root system develops

Page 2: The last two paragraphs of the Introduction seem out of place because they follow your statement of purpose (lines 58-59). A suggestion is to move lines 60-65 to line 53. I rearranged the order and edited some text.

Page 2: You need a final paragraph of the Introduction containing your hypothesis and objective. A suggestion is (1) Last paragraph stating with sentence at lines 68-69. (2) Next sentence is lines 66-68. (3) Next sentence is your hypothesis. (4) Last sentence is lines 58-59.  Final paragraph rearranged and stated hypothesis

You need to justify the use of cuttings as a surrogate for naturally seeded willow. Have done so in the final paragraph stated in the first and last paragraph of the introduction

Methods

Page 2, paragraph 1: More information is needed about the source of the cuttings.  For example, what was their source (e.g., native shoots from mature, naturally growing trees), what age were the trees at the time of cutting, and when were the cuttings made. Information now included

Page 2: You probably need an entire paragraph to describe the cuttings (see above).  Another paragraph is needed to describe the boxes.  The dimensions of the box are given, but you need to add dimensions of the rooting zone, and the three compartments serving as replications. This paragraph should contain information on how the rooting medium was created and how the PVC pipe was used.  I cannot follow how you centered the PVC pipe in each box, but there were three compartments in each box.  You need to provide more detail such as that the PVC pipe depth reached to bottom of the box.  Then start another paragraph with information on how the cuttings were planted (depth) in the PVC pipe and how and when the pipe was removed.  So there were only 5 cuttings used in the entire study? Methods now include more detail of procedure to cover these issues

Page 2: You mention that the boxes are near an irrigation system, but then you state that a sprinkler was used to water the cuttings.  It would be clearer if you used one term for these attributes. Reference now is only to overhead irrigation

Page 2, line 73: Is “root buds” a real term?  If not, try to use the technically correct term for this, perhaps “emerging root primordia”. Accepted advice and changed text accordingly

Page 2, line 85: Was the side of the box covered with transparent polycarbonate, or was an observation window created by replacing the wood on one side of the box with transparent polycarbonate? Clarified

Page 3, line 91: A new paragraph should begin with “Ten weeks…”  Statements like “layer by layer” are not appropriate in a scientific document. You need to do a better job of describing the depths of the “layers” earlier in the Methods.  Then at this point, make a statement indicating that all roots were extracted from each 7 cm layer substrate.  Language modified term used is ‘7 cm depth intervals’

Page 3, line 97: What sieve size? stated

Page 3, line 97: Samples of what… substrate, roots?  What was the volume of the 100 samples? Why wouldn’t you sample all roots? How does the reader know that the samples you chose were representative of those in the box?  Clarified. We had an issue with some of the samples being thrown out prematurely! Hence why we did not measure all samples.

Page 3, line 99: What was the basis of these diameter classes? Include a reference. Done

Page 3, line 100: Here you refer to “cell.” Are these the three compartments per box described earlier?  If so, use one term only. Was it 100 samples per cell or box? Compartment now used. !00 samples stated as being from all the boxes in total

Page 3, line 100: What is a “sample box”?  Is this the same as “box”? the word ‘sampl’e removed

Page 3:  How did you measure root length? Stated ‘using a ruler’

Page 3, lines 96-110: I cannot tell what was done with the harvested roots.  I can only guess at this point.  If I guess, then this paragraph should be separated into several with one each for (1) root harvest by layer, the tools used, root class identification, live/dead identification, now stated p3 line 109-110 (2) dry mass determinations of 100% of the root harvest by layer, how measured (nearest 0.1 g?), given 0.01 g (3) 100 subsamples (by what…volume, fresh weight) from each cell for RL determinations, how RL determined, (4) relationship development, equations by root class, (5) What RLD is and how is was calculated by cutting, (6) Shoot variables.

Page 3: There is no description of the experimental design, or how you did your mean separations (Figure 2).  As it stands, it looks like 5 cuttings were planted in each of five boxes. Each box had a specific substrate combination.  I don’t detect any replication except among the three cells per box. But, I’m not sure if this qualifies as replications since the observations came from the same cutting and box. This is not correct. There was replication with three cuttings per box, one in each compartment

Results

Page 3, line 113: “thin and thick roots” are not scientific terms.  You need to search the literature for standard descriptive terms that can be applied to your study.  You must define your root categories based on the literature. Changed to ‘fine’ and ‘coarse’. A reference is provided

Page 4, line 130: Are these cuttings or trees?  I don’t think they should be called trees. Why do you start with shoot description when root description is the emphasis of your paper? After 10 weeks the shoots were < 1.5 m. However I have changed ‘tree’ to ‘cutting’.

Page 4: ANOVA tables should be presented. I have retained ANOVA values in the text.

Page 4, line 131: There is no indication of how shoot number was quantified. What qualified as a shoot? Now defined p3 line123

Page 6-7: Bars in figures are too difficult to resolve. The figure could be enlarged if this is important.

Reviewer 2 Report

Authors are advised to carefully check the text for potential typing errors. Data in Table 2 should be indicated in 3 significant figures. Please, improve figures if possible.

Author Response

The authors checked the morphological traits along with the gradient of substrate composition in the riverbank ecosystem. I think it is important to gather the basic knowledge about how the willow can contribute to stabilize the substrate in the riverbank from the view point of the fine root traits. However, based on the present manuscript, it was difficult to judge the novelty of this study. Furthermore, I found many minor concerns as below.

First paragraph of Abstract: Please clarify which knowledge is lacking/little understood.

Line 13-14: Some missing words? Not understandable. Clarified

Line 15: Early stage of what? Not understandable. Early after the establishment of seedling on the riverbank? Added ‘grown from cuttings’

Line 17-: I am not sure whether the cutting is the proper way to test because the root properties from cutting is different from those from sprout.  Reason for using cuttings is explained in the introduction.

Line 26-27: This could be removed from the abstract because the main focus is the difference among substrates.

Line 37: Could be not the willow itself but the plantation of cutting of willow. The text has been changed here for better understanding of how willows are used for riverbank stabilisation in New Zealand.

Line 40-52: The description about insect damage is not closely linked to this study and redundant. Could be deleted. I have retained it for contextual reasons

Line 53-59: I understood that normal root development of tree willows growing along  riverbanks is scarce. However, the authors should demonstrate the related studies about, for example, 1) willow root plasticity for environmental gradient (even not in the river bank) and/or 2) the root morphology of non-willow species in river bank. Due to the lack of these references, I cannot really judge the novelty of this study. I have added references to willow roots studies in other substrate types, but have not extended the literature review to non-willow riverine species.

Statistical analysis: The authors have conducted ANOVA analysis. But to understand between which condition the difference existed, I think multiple comparison test such as Tukey HSD should be done (Figure 2-5 should include the statistical results). A fuller explanation of ANOVA analysis has been added to the methods in consultation with my statistician

Reviewer 3 Report

The authors checked the morphological traits along with the gradient of substrate composition in the riverbank ecosystem. I think it is important to gather the basic knowledge about how the willow can contribute to stabilize the substrate in the riverbank from the view point of the fine root traits. However, based on the present manuscript, it was difficult to judge the novelty of this study. Furthermore, I found many minor concerns as below.

First paragraph of Abstract: Please clarify which knowledge is lacking/little understood.

Line 13-14: Some missing words? Not understandable.

Line 15: Early stage of what? Not understandable. Early after the establishment of seedling on the riverbank?

Line 17-: I am not sure whether the cutting is the proper way to test because the root properties from cutting is different from those from sprout. 

Line 26-27: This could be removed from the abstract because the main focus is the difrence among substrates.

Line 37: Could be not the willow itself but the plantation of cutting of willow. 

Line 40-52: The description about insect damage is not closely linked to this study and redundant. Could be deleted.

Line 53-59: I understood that normal root development of tree willows growing along  riverbanks is scarce. However, the authors should demonstrate the related studies about, for example, 1) willow root plasticity for environmental gradient (even not in the river bank) and/or 2) the root morphology of non-willow species in river bank. Due to the lack of these references, I cannot really judge the novelty of this study. 

Statistical analysis: The authors have conducted ANOVA analysis. But to understand between which condition the difference existed, I think multiple comparison test such as Tukey HSD should be done (Figure 2-5 should include the statistical results).

Author Response

Yes

Can be improved

Must be improved

Not applicable

Does the introduction provide sufficient   background and include all relevant references?

(x)

( )

( )

( )

Is the research design appropriate?

(x)

( )

( )

( )

Are the methods adequately described?

(x)

( )

( )

( )

Are the results clearly presented?

(x)

( )

( )

( )

Are the conclusions supported by the results?

(x)

( )

( )

( )

Comments and Suggestions for Authors

Authors are advised to carefully check the text for potential typing errors. Data in Table 2 should be indicated in 3 significant figures. Please, improve figures if possible.

Recommended changes have been made. Table 2 data have been adjusted. Figures 3 and 5 will be improved by enlargement.